# Overcoming Asymmetric Contact Resistances in Al-Contacted Mg_2_(Si,Sn) Thermoelectric Legs

**DOI:** 10.3390/ma14226774

**Published:** 2021-11-10

**Authors:** Julia Camut, Sahar Ayachi, Gustavo Castillo-Hernández, Sungjin Park, Byungki Ryu, Sudong Park, Adina Frank, Christian Stiewe, Eckhard Müller, Johannes de Boor

**Affiliations:** 1Institute of Materials Research, German Aerospace Center (DLR), D–51170 Cologne, Germany; sahar.ayachi@dlr.de (S.A.); gustavo.castillo-hernandez@dlr.de (G.C.-H.); adina.frank@dlr.de (A.F.); christian.stiewe@dlr.de (C.S.); Eckhard.mueller@dlr.de (E.M.); 2Energy Conversion Research Center, Korea Electrotechnology Research Institute (KERI), 12, Jeongiui-gil, Seongsan-gu, Changwon-si 51543, Gyengsangnam-do, Korea; sjinpark@keri.re.kr (S.P.); byungkiryu@keri.re.kr (B.R.); john@keri.re.kr (S.P.); 3Institute of Inorganic and Analytical Chemistry, Justus Liebig University, D–35392 Giessen, Germany; 4Institute of Technology for Nanostructures (NST), Faculty of Engineering, University of Duisburg-Essen, D–47057 Duisburg, Germany; 5Center for Materials Research/LaMa, Justus Liebig University, D–35392 Giessen, Germany

**Keywords:** thermoelectrics, thermoelectric generator, contacting, contact resistance, electrode, silicides, process

## Abstract

Thermoelectric generators are a reliable and environmentally friendly source of electrical energy. A crucial step for their development is the maximization of their efficiency. The efficiency of a TEG is inversely related to its electrical contact resistance, which it is therefore essential to minimize. In this paper, we investigate the contacting of an Al electrode on Mg_2_(Si,Sn) thermoelectric material and find that samples can show highly asymmetric electrical contact resistivities on both sides of a leg (e.g., 10 µΩ·cm^2^ and 200 µΩ·cm^2^). Differential contacting experiments allow one to identify the oxide layer on the Al foil as well as the dicing of the pellets into legs are identified as the main origins of this behavior. In order to avoid any oxidation of the foil, a thin layer of Zn is sputtered after etching the Al surface; this method proves itself effective in keeping the contact resistivities of both interfaces equally low (<10 µΩ·cm^2^) after dicing. A slight gradient is observed in the n-type leg’s Seebeck coefficient after the contacting with the Zn-coated electrode and the role of Zn in this change is confirmed by comparing the experimental results to hybrid-density functional calculations of Zn point defects.

## 1. Introduction

With their ability to convert waste heat into electricity, thermoelectric generators (TEG) are a promising source of renewable energy. They have no moving parts, making their maintenance easy, and they have the advantage of being lightweight and reliable, which makes them relevant in fields such as the aerospace and the automotive industries [1,2,3].

A TEG is typically composed of n- and p-type thermoelectric (TE) legs, which are connected to metal-bonded ceramic plates electrically in series and thermally in parallel [4]. The legs are usually functionalized: they are contacted with a metallic layer, referred to as electrode, that will be then soldered to the metallic bridges, which interconnects the legs in an electrical circuit. This electrode can have a barrier diffusion function or be combined with another layer that plays this role, and is also applied to facilitate the bonding of the TE elements to the bridges. The establishment of a good electrical contact between metals, which are indeed simpler. Applying solders directly between the bridges and the TE legs could alter the TE properties due to diffusion processes. The design of the legs and the choice of the TE materials in TEGs can vary in order to suit their different applications [5].

To assess the performance of a TEG, its power output P and its conversion efficiency *η* can be calculated, using the TE material properties and the contact resistances between the TE legs and the electrodes. The equations for a TEG containing N pairs of thermoelectric legs are shown in the following equations [6]:(1)$$P=\frac{S^{2}\sigma }{2}\frac{NA{\left(T_{h}-T_{c}\right)}^{2}}{\left(l+n\right)}$$
(2)$$\eta =\left(\frac{T_{h}-T_{c}}{T_{h}}\right){\left\{2-\frac{1}{2}\left(\frac{T_{h}-T_{c}}{T_{h}}\right)+\frac{4}{zT_{h}}\left(\frac{l+n}{l}\right)\right\}}^{-1}$$
where *S* and *σ* are the Seebeck coefficient and electrical conductivity of a pair of n-type and p-type legs, *z* its figure of merit calculated as $z=\frac{S^{2}\sigma }{\kappa }$, where κ is the thermal conductivity. In this equation, equal properties for n- and p-type materials are assumed; in the case of unequal properties and cross-sections, the average of the Seebeck coefficients, of the electrical resistances and of the thermal conductances should be used. *N* is the number of leg pairs in the module; *A* is the cross-sectional area of a leg; *T_h_* and *T_c_* are, respectively, the temperatures at the hot and cold sides of the TE legs; l is the length of a TE leg; $n=2\sigma \rho_{c}$, where $\rho_{c}$ is the electrical specific contact resistance. The equations assume temperature independent properties and also neglect 2D or 3D effects.

These equations show that, for a given temperature range, material optimization and contact quality improvement are crucial to increase the TEG performance. Low electrical contact resistance (≤10% for each TE leg) and TE materials with a high figure of merit (zT ≥ 1) are standard requirements for industrial use of TEG [7].

The TE solid solutions Mg_2_Si_1-x_Sn_x_ (x ≈ 0.6–0.7) have reproducibly high thermoelectric properties with a figure of merit of up to zT = 0.6 for p-type and 1.4 for n-type at 450 °C [8,9,10,11,12,13,14]. On top of being lightweight, this material has the advantage of being inexpensive and non-toxic. Mg_2_X has a cubic anti-flourite Fm$\bar{3}$m crystal structure [15,16].

A few silicide-based modules combining n-type Mg_2_Si and p-type HMS (High-Manganese Silicide) legs were already reported [17,18,19,20,21,22,23,24]. However, the development of a fully Mg_2_(Si,Sn)-based TEG remains at an early stage. Gao reported a first attempt at building such a TEG [25], showing a maximum power output of 117 mW for *T_h_* = 440 °C and *T_c_* = 110 °C. Recently, Goyal et al. reported a power density of 0.52 W/cm^2^ and a computed predicted maximum efficiency of 5% for a fully Mg_2_(Si,Sn)-based TEG [26].

A first step in the development of a fully Mg_2_(Si,Sn)-based TEG is the electrode selection. Several attempts were reported with only partially promising results, as the electrodes provided low electrical contact resistance but also led to cracking (Ni, constantan) or altering of the TE material’s Seebeck coefficient (Cu, Ag) [27,28,29]. Most recently, aluminum was reported as the best electrode for this TE material system, showing low electrical contact resistance and good thermal and chemical stability of the interface also through annealing [30]. In that work, one of the reported samples showed asymmetric electrical contact resistances at the two contacting faces before annealing. The difference between the two sides was of several orders of magnitude and, although the highly resistive contact seemed to heal with annealing, the reason for this initial asymmetry could not be given. A similar case can also be found with silver electrodes [29], and we also encountered this issue with Ag and Ni electrodes, as shown in Appendix A. No similar phenomenon seems to have been reported in the thermoelectrics or even in the broader electronics field.

A high electrical contact resistance, even if only located on one side of the TE leg, is detrimental to a TEG performance. Even in the case of a reduction of the electrical contact resistance with annealing, which could happen during the lifetime of the TEG, this would create a consequent delay in optimal performance. A high electrical contact resistance could also potentially indicate a poor physical adhesion, which could lead to an early mechanical failure of a working device composed of such kind of legs. Since the technological steps towards TEG have not yet been thoroughly studied nor reported, it is likely that this asymmetry in contact resistances was or will be encountered in other studies.

This paper is focused on the study of this asymmetry phenomenon using aluminum electrodes, as previously reported. Based on the statistics of various samples, the influence of the typically most relevant processing parameters for contacting can be ruled out (e.g., temperature, pressure…). However, it was found that the asymmetry in contact resistances appears during the dicing step, as the high electrical contact resistance is systematically observed on the sample side, which is at the bottom, attached to the sample holder, during dicing. A strategy consisting in etching the Al foil and protecting it with a sputtered Zn layer as oxidation barrier prior to contacting is shown to effectively avoid the asymmetry in electrical contact resistances after dicing. This indicates that the rise in the bottom side electrode’s electrical resistivity during dicing is probably due to the presence of the native oxidation layer on Al foils prior to contacting. The added Zn layer does not trigger any chemical reaction detrimental to the interface adhesion, and does not alter the p-type Seebeck coefficient. A slight gradient in the Seebeck coefficient is observed for the n-type sample, but its magnitude is low enough to preserve future TEG’s performance. In this manuscript, we show an efficient way to overcome a technical challenge for the fabrication of a Mg_2_(Si,Sn)-based TEG.

## 2. Materials and Methods

### 2.1. Experimental

The solid solution Mg_2_(Si,Sn) pellets were prepared similarly to what was reported in previously published papers [27,30,31,32], with the following nominal stoichiometry: Mg_2.06_Si_0.3_Sn_0.665_Bi_0.035_ for n-type and Mg_1.98_Li_0.03_Si_0.3_Sn_0.7_ for p-type. The n-type composition contains Mg with an excess of 3% in order to compensate for the Mg evaporation occurring during the sintering, due to its longer duration compared to p-type samples. The pellets’ preparation prior to contact as well as the contacting parameters are also identical to what was previously published [30]. Unless specified otherwise, the contacting temperature for the samples presented in this paper is 475 °C. After contacting, the pellets are diced into legs using a Disco DAD321 Automatic Dicing Saw). The cutting speed through the sample was 0.3 mm/s with an angular speed of 30,000 blade rotations per minute and each cut was made in two passes, keeping the mechanical stress to the sample by the blade´s displacement as low as possible.

The quality of the joining is estimated by the value of its specific contact resistance rc and the preservation of the Seebeck coefficient of the TE material using a Potential & Seebeck Scanning Microprobe (PSM) [33]; and by its microstructural and chemical composition at the interface using scanning electron microscopy (Zeiss Ultra 55 SEM equipped with an EDX detector). Two $r_{c}$ values are obtained for each contact with two different calculation methods, using the TE material´s electrical conductivity to calculate the current density ($r_{c,j\left(TE\right)}$), or using the current passing through the sample as measured by the PSM ($r_{c,j\left(PSM\right)}$), as reported in previously published papers [27,30]. The two specific electrical contact resistances are calculated using the following equations:(3)$$r_{c,j\left(TE\right)}=\frac{\left(V_{elec}-V_{TE}\right)\times l_{TE}}{\Delta V_{TE}\times \sigma_{TE}}$$
(4)$$r_{c,j\left(PSM\right)}=\frac{\left(V_{elec}-V_{TE}\right)\times A}{I_{PSM}}$$
where $V_{elec}$ is the potential on the electrode (metallic) at the interface and $V_{TE}$ is the potential on the TE material at the interface, the position of interface being located using the drop in Seebeck coefficient on the line-scan. The difference $V_{elec}-V_{TE}$ corresponds, therefore, to the drop of potential across the electrode-TE interface. $l_{TE}$ is the length of the TE material (between the two electrodes), $\Delta V_{TE}$ is the drop of potential across the TE material and $\sigma_{TE}$ is the electrical conductivity of the TE material, measured using a 4-probe inline technique, A is the sample cross-section and $I_{PSM}$ the current measured in the device.

Multiple line-scans are measured for each sample, each $r_{c}$ value presented below is the average value of all the lines, given with the corresponding standard deviation.

The calculation of the electrical contact resistance according to Equation (4) requires the assumption that the current density is homogeneous over the whole sample (along cross-section and length), while Equation (3) only assumes a constant current density along the direction of the line-scan. The more inhomogeneous the interface, the more different are the results given by both equations. The comparison of those two values allows to assess the reliability of the measurement and the quality of the Al/TE interface.

The Zn coating suggested as an oxidation barrier was applied by magnetron sputtering using an AXPLORER 4375 S3E1 PVD machine. An ion etching process was performed on the Al foil surface prior to the Zn coating in order to remove the oxide layer on the Al surface. The coating was deposited at room temperature on a substrate rotating at 10 min^−1^ with a starting vacuum of 5 × 10^–6^ mbar. The duration of the coating was 90 min with a growth rate of 1.3 nm/s. The Zn target used for this process was provided by EvoChem (purity 99.99%).

### 2.2. Computational Method

First-principles calculations were performed to study the defect stability of Zn related point defects in Mg_2_Si and Mg_2_Sn using hybrid-density functional calculations [34]. The planewave basis set and the projector-augmented wave pseudopotentials [35] were used, as implemented in the Vienna Ab initio Simulation Package (VASP) code [36]. For the exchange-correlation functional, the Perdew–Burke–Ernzerhof parameterized generalized gradient approximation [37] was used. For hybrid calculations, the HSE06 is used with the exact mixing fraction of 25% and the screening parameter of 0.208 Å^−1^ [34].

For the defect stability, the charged defect formation energy of defect *D* with charge state *q* (*D^q^*), where *q* is +2, +1, 0, −1, −2, was calculated using the following equation [38]
(5)$$E_{\mathrm{Form}}\left[D^{q}\right]=E_{\mathrm{tot}}\left[D^{q}\right]-E_{0}-\Sigma_{i}\;\left(\;\mu_{i}\;\delta n_{i}\right)+q\left(E_{F}\right)$$
where $E_{\mathrm{tot}}\left[D^{q}\right]$ and $E_{0}$ are total energies with and without defects, subscript i indicates the atomic element, $\mu_{i}$ is the atomic chemical potential, $\delta n_{i}$ is the change of number of i-th element in the defective supercell with respect to the pristine one, and $E_{F}$ is the Fermi level of the system. Detailed information can be found in our previous work [39,40,41].

The defect density of $D^{q}$ in Mg_2_Si or Mg_2_Sn can be estimated using the defect formation energy and the Boltzmann factor for a given material synthesis temperature.

## 3. Results

Figure 1 shows the ratio of the electrical contact resistance of both sides of the functionalized leg. It displays a quite large range of $r_{c1}/r_{c2}$ (factor of 1 to 100) extracted from more than 30 Al-contacted samples and various combinations of experimental parameters. It can be seen that a minority of the samples actually have symmetric contact resistances, such as those shown in the study about aluminum contacting [30]. Some experimental factors, such as contacting temperature, pressure and TE pellet length, were suspected as more likely to be the causes for the asymmetry. However, when plotted accordingly, none of them shows a significant correlation with the asymmetry, as can be seen in Figure 1.

Other factors—carrier type, TE pellet geometry (diameter of 15 or 30 mm), presence or absence of a buffer layer during contacting, direct vs. indirect setup—did not show a clear correlation either, see Appendix A. Indeed, asymmetrically contacted samples could be randomly found for all “configurations” of those factors.

In order to determine if this asymmetry phenomenon is linked to the joining or the dicing step in the leg fabrication process, two differential experiments were designed. In the first one, the side of the sample which is at the top during the contacting step is also on the top during the dicing step, while in the second one, the sides are switched (the top side during contacting goes at the bottom during dicing). The experiments are labelled “X/Y” with X being the position of one side during contacting and Y the position of that same side during dicing. After contacting, the pellets are cut in 9 square-based legs. The results of the top/top experiment are shown in Table 1**.** while the results of the top/bottom experiment are shown in Table 2. The two contact resistivity values are obtained using the Equations (3) and (4). Exemplary line scans of the potential and the Seebeck coefficient are shown in Appendix A for each experiment.

The most important observation from both tables is that the side with the higher specific contact resistance is always the side at the bottom during cutting, which indicates that the asymmetry is not linked to the contacting but to the dicing step.

It can also be seen that some legs have a very high standard deviation. This typically indicates non-uniform interfaces in a sample: as the potential drop at the interface is greatly varying line to line due to varying contact quality, the resulting variation in the contact resistivity for one sample is higher. The asymmetry is generally more pronounced (higher $r_{c}$ values at the bottom during dicing) for the top/top sample (Table 1) than for the top/bottom sample (Table 2). It is seen for both kinds of samples that the magnitude of the low $r_{c}$ does not vary a lot, as it mostly remains below 15 µΩ·cm^2^, while high $r_{c}$ values vary more leg to leg.

In Appendix A, EDX line scans of the Al/TE interface are presented. It can be seen that for higher $r_{c}$, a peak of oxygen is observed at the junction. This peak is much weaker for interfaces with lower $r_{c}$, indicating a correlation between the initial presence or absence of oxidation on the Al foil and the magnitude of the specific electrical contact resistance.

Aluminum is known to oxidize quickly and to form a very stable oxide, Al_2_O_3_. Before joining, the Al foils are polished with SiC paper in order to diminish the thickness of the oxide layer. As this is a manual process, its reproducibility is limited (wear of the paper due to prior grinding of other foils, varying exposure time to air before the start of the joining process). This method also does not completely prevent the presence of an oxide layer, as a new, fresh Al_2_O_3_ layer forms within less than seconds under air. It is therefore suspected that the presence of oxide prior to contacting could play a role in the appearance of the high contact resistance during dicing, where the pulling forces would damage those areas with weaker adhesion.

To test this hypothesis, the Al foils were ion-etched under Ar in the PVD, with subsequent sputtering of an 8 µm Zn layer as oxidation protection. SEM pictures of the coating and a picture of a contacted pellet are shown in Appendix A. After Al/TE contacting with the coated foils, the functionalized pellets were cut similarly to the previous samples. The specific contact resistances for all legs are reported in Table 3. Zn was chosen due to its known solubility in Al [42,43]. Zn can still oxidize under air, which is why the foils are stored under Ar before contacting. However, a thickness of 8 µm is enough to protect the Al for at least the time necessary until the contacting process. Moreover, Zn has a low melting point which allows it to act as a solder during contacting and to break the oxide layer which might have formed at its surface.

After dicing, most legs have low and symmetric $r_{c}$, which indicates that removing oxidation from the Al foil is an effective strategy to avoid asymmetric electrical contact resistances. A possible explanation for the asymmetry behavior could therefore be that local oxidation spots are present at the interface. During cutting, the interface around those points could be pulled apart more easily due to a concentration of stress, which would increase $r_{c}$.

An SEM picture of the interface, the corresponding EDX points analysis as well as an EDX mapping are respectively reported in Figure 2, Table 4 and Figure 3.

It was previously shown that there was no reaction at the interface between Al and Mg_2_(Si,Sn) directly after contacting [30]. Here, with the Zn coating, the interconnection zone contains scarce small areas with newly formed phases, of a diameter of 20–40 µm. They are mostly composed of a ternary alloy (Zn; Mg; Al) containing pure Al dots. At the interface between the interconnection zone and the Mg_2_(Si,Sn) matrix, a very bright Bi-rich phase is formed. From the mapping, it can be seen that, similarly to experiments with uncoated Al foils [30], there is no composition gradient neither in Al electrode, nor in the TE material next to the nuclei.

Contacting an electrode with a Zn layer implies that some Zn could diffuse into the TE material. This can potentially have detrimental effects on the TE properties (and potentially also on the thermal stability of the material) as was already observed with other electrodes such as Ag and Cu [29,32,39]. In order to check for those effects, Seebeck coefficient measurements were performed and reported in Figure 4. The results are from measurements after sintering (in black) and after contacting (in red).

For the p-type sample, the Seebeck coefficient changes by less than 10% between the sintered and the contacted states, which is negligible and could be attributed to the wear of the tip of the PSM. For the n-type sample, a gradient in the Seebeck coefficient appears after contacting. There is a difference of about 20 µV/K from the TE material-electrode interface to the center of the n-type material (−110 µV/K at the interface, −90 µV/K in the middle). The appearance of a gradient after contacting is similar to what was reported on contacting with Ag electrodes, in which case the gradient was explained by charge carrier compensation due to Ag diffusion into the TE material [29,39]. This hypothesis was supported by the relatively low formation energies of electrode-related point defects compared to the intrinsic and dopant-induced point defects.

In order to verify if the same mechanism can explain our experimental results, hybrid-DFT calculations were performed to study the defect stability of Zn-related defects in Mg_2_Si and Mg_2_Sn. The possible formation of Zn-related point defects is confirmed by comparing the formation energies of the Zn-related point defects to those of intrinsic and extrinsic (Bi and Li) defects from previous papers [39,40]. As the samples of interest presented in this paper are Sn-rich Mg_2_(Si,Sn) solid solutions, only the results of Mg_2_Sn are shown in Figure 5, while the results for Mg_2_Si are shown in Appendix A. In these calculations, both Mg-poor and Mg-rich chemical potential environments were considered. In fact, the experimental n-type samples were synthesized with 3 at.% excess Mg, which means an initially Mg rich environment. However, after sintering and joining, Mg loss is very probable due to Mg evaporation and Mg diffusion. Therefore, it makes sense to also investigate the defect formation energies under Mg poor conditions. The case of p-type Li-doped Mg_2_Sn under Mg-rich conditions will not be discussed. Indeed, Li doping is aimed at Mg sites in the Mg_2_X material; however, it was found that, although Li on Mg site defects are stable, Li interstitials are even more stable defects [39]. This means that the Li-doped material will always contain a certain proportion of Mg vacancies and Mg-rich conditions will not be obtained. For simplicity, only the most stable defects (with E_form_ < 1 eV) are presented in the figure below and discussed. Note that if the defect formation energy difference is larger than 1 eV, the defect number density is negligible due to its exponential dependence on the defect formation energy [40].

Figure 5 shows the charged defect formation energy as a function of the Fermi level for various defects in the case of Zn diffusion in Bi- and Li-doped Mg_2_Sn. Here, we considered the Zn interstitial (Zn_int, or I_Zn_), Zn substitutional at Mg (Zn_Mg or Zn_Mg_), and Zn substitutional at Sn site (Zn_Sn or Zn_Sn_).

Figure 5a,b show the defect formation energies of Zn in p- and n-type, respectively, Mg-poor Mg_2_Sn. It is seen that in the p-type material, the Li_Mg_ (Li_Mg) defect (p-type dopant) is more stable than the most stable defect brought by Zn, meaning that no charge compensation is to be expected. For the n-type, the Zn_Mg_ defect is the most stable Zn-defect in Sn-rich Mg_2_Sn and the corresponding defect formation energy is only 0.25 eV. As Zn and Mg have the same number of valence electrons, the Zn_Mg_ is an isovalent defect, which does not donate any hole or electron carrier if the Fermi level is at the middle of the bandgap. However, when the Fermi level is at the CBM, the charge state of Zn_Mg_ can be negative, indicating a possible electron trapping by Zn_Mg_. This can be explained by the fact that, although Zn is isovalent to Mg, it can attract electrons as it has higher electronegativity compared to Mg. Similarly, conduction band subgap states acting as electron trap induced by metal cation disorder or anion isovalent doping were reported for oxide semiconductors [44] and GaAs [45].

Under Mg-rich conditions, Zn_Sn_ is the most stable Zn-related point defect and its charge state is always negative over the whole *E*_F_ range inside the band gap, as shown in Figure 5c.

From the above-charged defect formation energies, we elucidate that the Zn-related defects in Mg_2_Sn can be electron trap centers in n-type Mg_2_Sn. Ayachi et al., reported that Ag or Mg-related native defects diffuse easily in Mg_2_Si and Mg_2_Sn [39]. Similarly, we may expect that the Zn in the surface of Zn-coated Al electrode can diffuse towards Mg_2_Sn. The Zn_Sn_ and Zn_Mg_ double electron killers can be generated inside the Mg-rich or Sn-rich n-type Mg_2_Sn, respectively. As they can easily trap the electrons in the n-type Mg_2_Sn where *E*_F_ is near the CBM, a charge compensation would occur.

These predictions of the hybrid-DFT calculations match with the experimental results obtained in Figure 4. Indeed, we see no change in the p-type carrier concentration and a decrease of n-type carrier concentration close to the contacts, indicated by the Seebeck coefficient as S is inversely related to n. Diffusion of the sputtered Zn is therefore strongly suspected to be the origin of the change in TE properties.

## 4. Discussion

Although Al was found to be a promising electrode for the Mg_2_(Si,Sn) material system [30], the phenomena of asymmetry in electrical contact resistances on a functionalized leg was repeatedly observed on multiple Mg_2_(Si,Sn) samples contact to Al electrodes with various experimental parameters. No correlation could be found between the ratio of electrical contact resistivities and the contacting parameters. Furthermore, the contacting-dicing experiment of this work showed that the high $r_{c}$ was systematically found at the bottom side from the dicing step, which indicates that the asymmetry does not originate from the contacting but from the dicing process.

The presence of an oxygen peak on the EDX line scans at the Al/TE interfaces with high electrical contact resistances (Appendix A), as well as the disappearance of the asymmetry when oxidation is avoided on the Al foil, indicates that the native oxide layer of the Al electrode could be the origin of the phenomenon. The oxide can prevent proper adhesion [46], which creates areas where the contact between the TE material and Al is weak and more sensitive to the mechanical stress applied during dicing [47].

Using ion-etching and Zn sputtering as an oxidation barrier is shown to be an efficient method to ensure low and symmetric electrical contact resistivities after dicing. A gradient in the Seebeck coefficient is observed after contacting on the n-type sample, indicating a decrease in carrier concentration in the TE material close to the interfaces. This behavior shows that the diffusion of one or several elements changes the defect concentrations, which alters the local carrier concentration. Given the low magnitude of this gradient, the corresponding change in carrier concentration should be minor and there should not be any significant impact on the performance of the leg in a TEG. Indeed, supposing that the whole leg’s Seebeck coefficient changed from −90 µV/K to −110 µV/K (instead of only the areas close to the contacts), it can be deducted, using the Pisarenko plot in Appendix A, that the carrier concentration would change from n= 3.1 × 10^26^ m^−3^ to n= 2.2 × 10^26^ m^−3^. It is known that for many material systems, including Mg_2_(Si,Sn) solid solutions, the zT, and therefore the efficiency, are not so sensitive to such a change in carrier concentration, as its peak extends on a broad n range [48].

The formation of small islands with new phases at the interface shows an affinity between Zn, Al, Mg and Bi, as a new (Mg;Al;Zn) phase is formed and Bi is segregated at the edge of this new phase. This might indicate that the change in carrier concentration may not, as discussed above, be due to the creation of Zn-related defects in Mg_2_(Si,Sn), but rather to the diffusion of Mg or Bi out of Mg_2_(Si,Sn). With the employed microstructural techniques, no gradient for one of the elements is discernable, we can therefore not differentiate between both options nor exclude a combination of both.

The case of Al diffusion into n-type Mg_2_(Si,Sn) was already discussed in a previous work, where no gradient was found after contacting with non-coated Al foils [30], similarly to the first results in this work (see line scans in Appendix A). It also appears that Al is a poor dopant for this material system as shown in [49,50,51,52,53]. It is, therefore, unlikely that the diffusion of Al into the n-type TE material is at the origin of the change in carrier concentration.

The effect of Zn is investigated by hybrid-DFT calculations, which predicts no change in carrier concentration of the Li-doped p-type material and a probable change of the Bi-doped n-type material, due to quite stable Zn_Mg_ and Zn_Sn_ defects which are electron killers. In literature, the effect of Zn on n-type Mg_2_(Si,Sn) was studied by Andersen et al., who also reported an increase in absolute n-type Seebeck coefficient in Zn-doped Mg_2_Si_0.4_Sn_0.6_ [54]. Although this does not completely dismiss the effect of the other elements suggested above, it suggests the clear and major role of Zn in the gradient observed in the samples with Zn-coated Al electrodes.

It was mentioned above that the gradient in Seebeck coefficient due to Zn diffusion is light and not detrimental to a TEG performance. It would, however, be possible to avoid this gradient altogether by finding a different oxidation barrier that would not create stable defects in the n-type material, using DFT calculations for the barrier selection [39]. Nevertheless, a far more important point to ensure long-term TEG performance is the material stability. It has been already shown that the n-type TE properties are very sensitive to Mg evaporation, which greatly alters the carrier concentration [27,30,55,56,57]. The investigation of suitable coating and annealing conditions should therefore be prioritized for further TEG development, see, e.g., Refs. [58,59,60,61,62,63].

## 5. Conclusions

We reported a technological difficulty related to the making of functionalized thermoelectric legs: asymmetry in electrical contact resistances. This issue could be commonly encountered in the field of thermoelectric materials in the present and future, as it appears during a widely used preparation process (pellet metallization followed by leg dicing). We investigated its origins and searched for an efficient solution to avoid it. It is found that it is crucial to avoid the presence of oxidation on the foil to be contacted as electrode, as this will tend to cause asymmetric contacts during the dicing step by the mechanical stress applied to the electrode/TE interface. An effective way for this is to ion-etch the metallic foil and coat the oxide-free surface with an oxidation protecting layer. In this work, the method was tested with Zn as oxidation barrier for Al on Mg_2_(Si,Sn). It successfully suppressed the contact asymmetry, allowing to reach low specific contact resistances on both sides of the samples (<10 µΩ·cm^2^). A slight gradient in n-type Seebeck coefficient is observed close to the Al/TE interfaces due to the Zn diffusion, which should not be detrimental to the leg performance in a TEG. In this study, an efficient method to ensure low and symmetric electrical contact resistivities between Al and Mg_2_(Si,Sn) was found. It may also be transferable to other material systems, thus making one more step towards TEG development.

## Figures and Tables

**Figure 1 materials-14-06774-f001:**
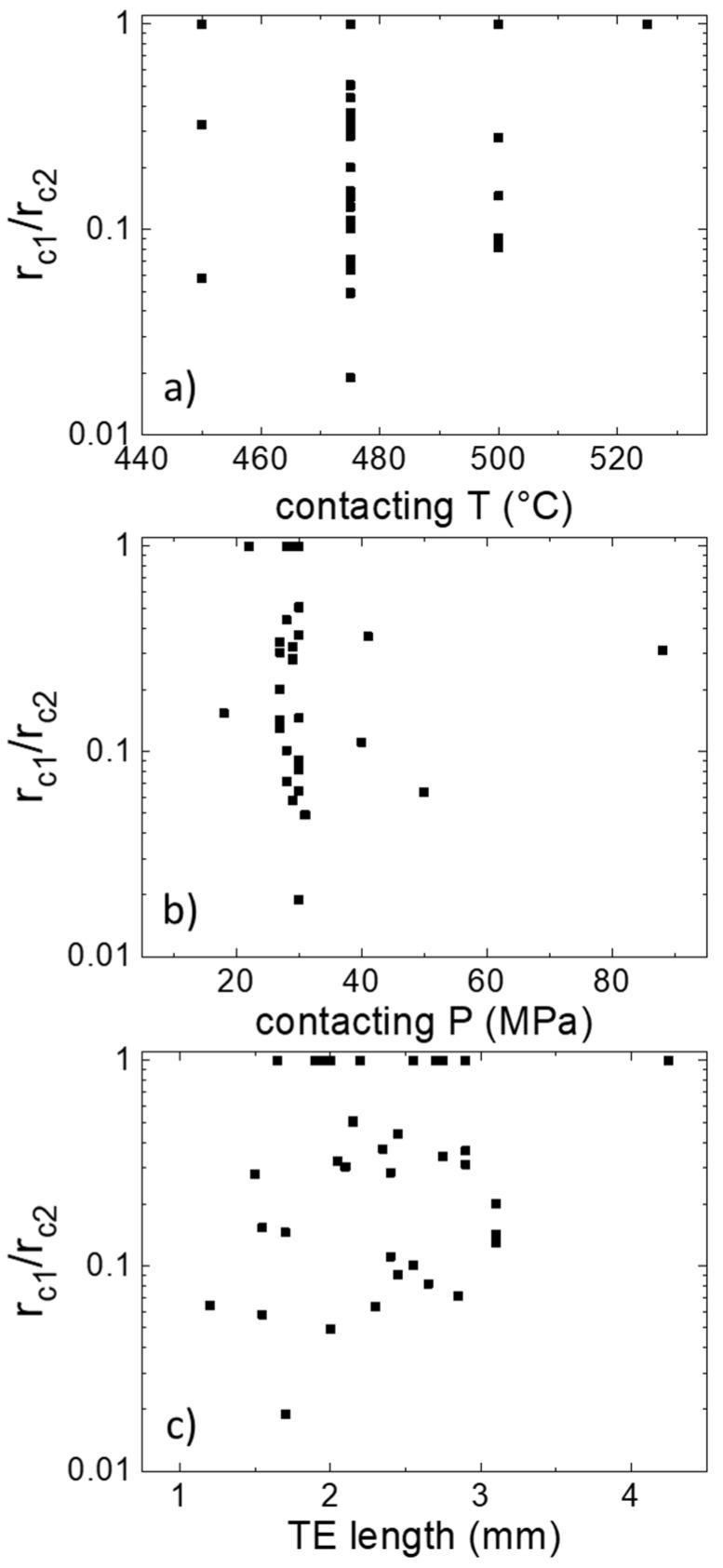
Ratio of the two electrical contact resistances of the same sample (smaller/larger) depending on various experimental parameters: (**a**) contacting temperature, (**b**) contacting pressure, (**c**) TE length before contacting. The samples are all Mg_2_(Si,Sn) with Al electrodes.

**Figure 2 materials-14-06774-f002:**
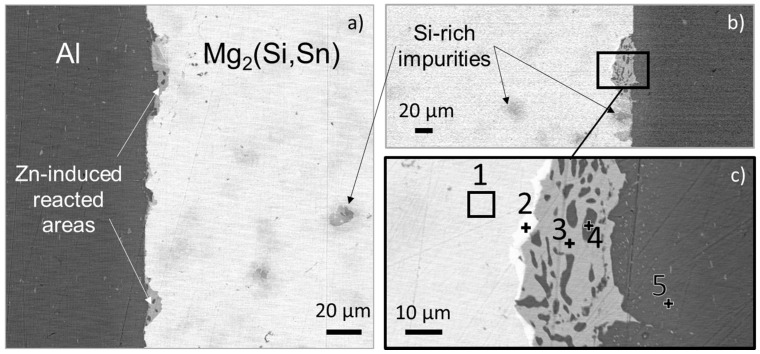
SEM images of different areas of the interface after contacting of a Zn-coated Al foil with a Mg_2_(Si,Sn) pellet: (**a**) first side interface, (**b**) second side interface, (**c**) zoom-in and point analysis of the reacted area in (**b**).

**Figure 3 materials-14-06774-f003:**
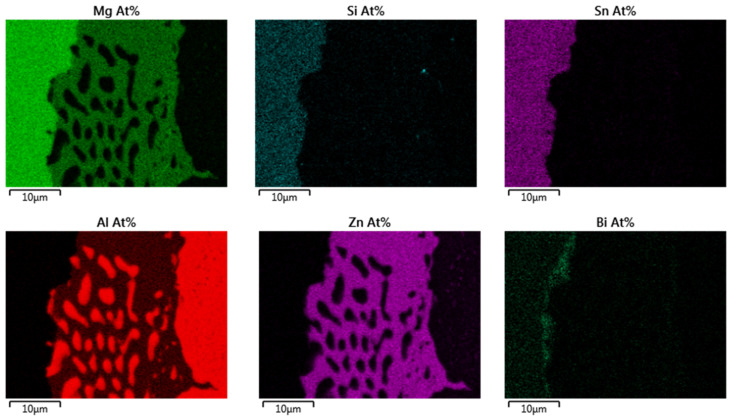
EDX mapping of a reaction area between Al, Zn and the Mg_2_(Si,Sn) TE material.

**Figure 4 materials-14-06774-f004:**
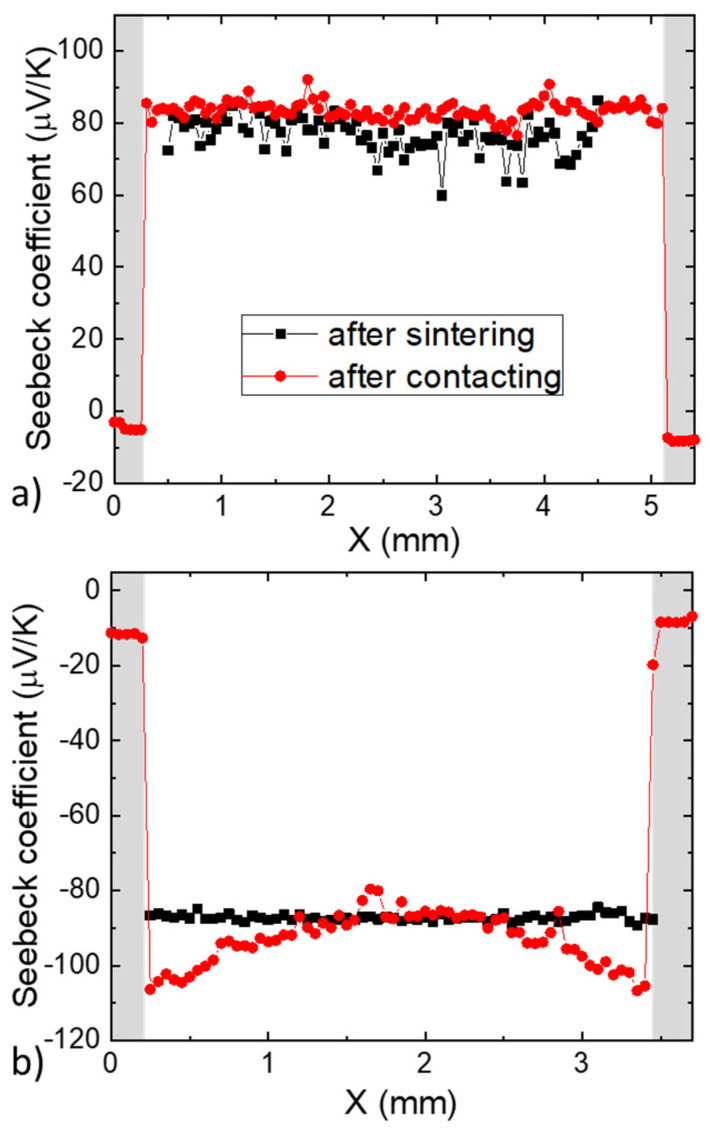
Seebeck scans along the (**a**) p-type, (**b**) n-type Mg_2_(Si,Sn) pellet contacted with Zn-coated Al foils after sintering and directly after contacting at 475 °C.

**Figure 5 materials-14-06774-f005:**
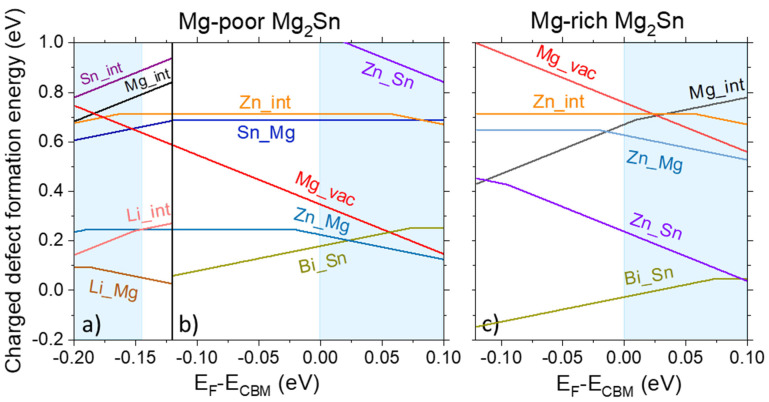
Charged defect formation energies for Zn defects in: (**a**) Li-doped (**b**) Bi-doped Mg_2_Sn under Mg-poor conditions (**c**) Bi-doped Mg_2_Sn under Mg-rich conditions. For simplicity, the results relevant for the Li-doped samples are represented in the Fermi level region around VBM only.

**Table 1 materials-14-06774-t001:** Specific electrical contact resistances $r_{c}$ on both sides of several legs, diced from a pellet contacted with Al foils. Leg 6 is missing as it broke during preparation, and leg 1 was cracked, which led to a massive potential drop, therefore the corresponding values are not reported.

Top/Top		Leg 2	Leg 3	Leg 4	Leg 5	Leg 7	Leg 8	Leg 9	Average
Side on top during dicing	$r_{c,j\left(TE\right)}$ (µΩ·cm^2^)	12 ± 8	8 ± 5	5 ± 2	9 ± 5	25 ± 13	7 ± 4	9 ± 4	10.7
	$r_{c,j\left(PSM\right)}$ (µΩ·cm^2^)	14 ± 9	7 ± 4	9 ± 5	11 ± 6	18 ± 9	5 ± 3	10 ± 4	10.6
Side on bottom during dicing (µΩ·cm^2^)	$r_{c,j\left(TE\right)}$ (µΩ·cm^2^)	368 ± 246	380 ± 57	702 ± 90	256 ± 100	305 ± 19	396 ± 80	298 ± 50	386.4
	$r_{c,j\left(PSM\right)}$ (µΩ·cm^2^)	417 ± 278	301 ± 45	1435 ± 184	306 ± 120	218 ± 13	313 ± 64	350 ± 59	477.1

**Table 2 materials-14-06774-t002:** Specific electrical contact resistances $r_{c}$ on both sides of a pellet contacted with Al foils, after dicing into several legs. legs 1, 2 and 3 broke during grinding. Legs 4 and 6 got detached from the sample holder and are therefore not identifiable.

Top/Bottom		Leg 5	Leg 7	Leg 8	Leg 9	Average
Side on top during dicing	$r_{c,j\left(TE\right)}$ (µΩ·cm^2^)	7 ± 5	15 ± 7	6 ± 2	7 ± 5	7.5
	$r_{c,j\left(PSM\right)}$ (µΩ·cm^2^)	8 ± 6	12 ± 5	7 ± 3	9 ± 6	8.8
Side on bottom during dicing (µΩ·cm^2^)	$r_{c,j\left(TE\right)}$ (µΩ·cm^2^)	581 ± 101	199 ± 22	100 ± 46	107 ± 25	258.2
	$r_{c,j\left(PSM\right)}$ (µΩ·cm^2^)	699 ± 122	156 ± 17	115 ± 53	144 ± 34	317.7

**Table 3 materials-14-06774-t003:** Specific electrical contact resistances $r_{c}$ on both sides of legs cut from a pellet contacted with Zn-coated Al foils. No leg from this sample broke.

Zn-Sputtering		Leg 1	Leg 2	Leg 3	Leg 4	Leg 5	Leg 6	Leg 7	Leg 8	Leg 9	Average
Side on top during dicing	$r_{c,j\left(TE\right)}$ (µΩ·cm^2^)	3 ± 1	5 ± 4	3 ± 1	3 ± 1	7 ± 7	3 ± 1	2 ± 1	3 ± 1	4 ± 2	3.7
	$r_{c,j\left(PSM\right)}$ (µΩ·cm^2^)	3 ± 1	7 ± 6	3 ± 2	5 ± 1	15 ± 14	4 ± 1	2 ± 1	4 ± 1	7 ± 5	5.6
Side on bottom during dicing (µΩ·cm^2^)	$r_{c,j\left(TE\right)}$ (µΩ·cm^2^)	10 ± 7	12 ± 4	3 ± 1	3 ± 1	9 ± 5	3 ± 1	4 ± 1	46 ± 8	3 ± 2	10.3
	$r_{c,j\left(PSM\right)}$ (µΩ·cm^2^)	20 ± 12	18 ± 7	4 ± 2	5 ± 2	19 ± 9	4 ± 1	5 ± 1	59 ± 10	6 ± 3	15.6

**Table 4 materials-14-06774-t004:** EDX analysis of the points indicated in Figure 2. The quantities are given in at%.

Point	%Mg	%Si	%Sn	%Bi	%Zn	%Al
1	64.4	8.8	25.6	1.2	-	-
2	63.6	9.0	21.7	5.7	-	-
3	33.2	-	-	-	41.9	24.9
4	2.3	-	-	-	3.4	94.2
5	1.5	-	-	-	3.6	94.9

## Data Availability

The data presented in this study are openly available in FigShare at [10.6084/m9.figshare.16751998].

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
