# Peer review of "Overcoming Asymmetric Contact Resistances in Al-Contacted Mg2(Si,Sn) Thermoelectric Legs"

_materials, 2021, doi:10.3390/ma14226774_

Round 1

Reviewer 1 Report

It is a very nice manuscript of significant practical interest. It definitely will be useful for researchers and engineers working in the field of thermoelectric generators design and development.

It can be accepted after correction of the following misprints:

Line 19. should be "...allow one to identify..."

Line 52, 53. the formula should end with a comma, next line should start with a small letter, no indent

Line 211. remove indent

Line 236. remove indent

Line 455. broken link

Remove table between lines 208-209 and 233-234

Author Response

We apologize to the reviewers. It seems that the conversion from the Word file to PDF format lead to dynamic link breakings (line 455) and duplication of floating tables (lines 208-209 and 233-234). These format issues have been corrected and the exportation to PDF should run smoothly again.

The indents (lines 53, 211, 236) are part of the template formatting provided by the journal Materials, therefore we do not feel comfortable deleting them and think this should be modified or not by the editor. Similarly, the suggestion of adding commas after equations (1) and (2) was not followed as the equations are on single lines and not included in the text. As a consequence, adding commas do not seem to fit the traditional formatting.

Line 19 has been corrected as suggested by the reviewer.

The capital letter line 53 has been replaced like suggested.

We would like to thank the reviewer for all the help and feedback.

Reviewer 2 Report

I propose that the authors briefly characterize Mg2Si crystal structure. What is a  structure of the Mg2Si1-xSnx solid solution within the range of x=0.6-0.7 ? Have the authors considered XPS studies of oxidized surface of foil before its ion etch ? What is the thickness of oxide layer ? 

In my opinion, the work is important and worth publishing.

Author Response

We would like to thank the reviewer for the feedback.

On the crystal structure: Mg2X has a cubic anti-flourite Fmm crystal structure with the Mg atoms occupying the eight tetrahedral positions (8c) and the X atoms at the corners and face centers (Wyckoff positions) [1]. Our material is synthesized following a reproducible process which was reported in literature [2] (Supplementary Information). The XRD pattern can be found page 4 here. An XRD characterization should therefore not be necessary in this paper, as the same synthesis process was used.

Line 69, the following text has been added: “Mg2X has a cubic anti-flourite Fm3 ̅m crystal structure [15, 16].”

On the oxide layer on Al: We would like to thank the reviewer for this suggestion. A more detailed analysis of the contact resistances on a nm-scale is planned in the future and XPS could be useful to clarify the role of oxide. Unfortunately, we do not have access to XPS characterization, and the foil and oxidation layer would be to thin to be characterized using XRD. The following paper investigated native oxide foil of Al and found a thickness of 2.7 nm with XPS: [3]. In our paper, we do not have direct proof of the role of the oxide layer, but an indirect observation of the effect of etching (lack of oxide layer), which leads to a dramatic decrease of the electrical contact resistance.

  1. Liu, X., et al., Significant roles of intrinsic point defects in Mg2X (X= Si, Ge, Sn) thermoelectric materials. Advanced Electronic Materials, 2016. 2(2): p. 1500284.
  2. Farahi, N., et al., High efficiency Mg 2 (Si, Sn)-based thermoelectric materials: scale-up synthesis, functional homogeneity, and thermal stability. RSC Advances, 2019. 9(40): p. 23021-23028.
  3. Evertsson, J., et al., The thickness of native oxides on aluminum alloys and single crystals. Applied Surface Science, 2015. 349: p. 826-832.

Reviewer 3 Report

This manuscript by Camut et al reports the asymmetry in contact resistances of Al-electrodes of Mg2 (Si,Sn) thermoelectric legs and then, they propose a novelty strategy to overcome that asymmetry. The topic has interest in the field, but the manuscript is careless presented, so, I recommend to publish it in Materials after minor revisions. 
My comments and questions are listed below:
1.    In line 70, HMS should be defined.
2.    In line 137, the equation (3) …is cancelled or why it shows red marks?
3.    In lines 140 and 141, V(elec) and V(TE) potentials are not properly defined. At a first glance, both appear to be the same potential at the interface.
4.    Maybe line 207 follows in line 212…isn’t it?
5.    Line 219, Table is repeated.
6.    Line 225. The data of Legs 4 and 6 are very confusing, I recommend remove those values. The high variation in standard deviation should be clarified.
7.    Line 245, maybe the action to remove the oxide layer with SiC  paper should be defined as “polish” instead “grind”.
8.    Line 255, Figure S8SI is not find in the supplementary information section, maybe it is a mistake.
9.    In line 455 of main text and in Supplementary Information section section, the “Error! Reference source not found” appear. This formatting mistake should be fixed.

After my general comments, I also recommend a carefully revision of the manuscript presentation.

Author Response

We apologize to the reviewers. It seems that the conversion from the Word file to PDF format lead to numbers of dynamic link breakings and duplication of floating tables. These format issues have been corrected and the exportation to PDF should run smoothly again.

Line 70: HMS was better defined: High-Manganese Silicide

The red markings on equation (3) pointed out by the reviewer do not appear on the files that we obtained after reviewing on the journal's website. We supposed that this was due to a version error.

Lines 140-141 were modified as follows (italic text was added): “Where  is the potential on the electrode (metallic) at the interface and  is the potential on the TE material at the interface, the position of interface being located using the drop in Seebeck coefficient on the line-scan. The difference  corresponds therefore to the drop of potential across the electrode-TE interface.”

Line 225, legs 4/6 have been removed from the table as suggested. The text below the table was modified as follows: “It can also be seen that some legs have a very high standard deviation. This typically indicates non-uniform interfaces in a sample: as the potential drop at the interface is greatly varying line to line due to varying contact quality, the resulting variation in the contact resistivity for one sample is higher”.

Line 245, “grinded” was replaced by “polished” as suggested.

We would like to thank the reviewer for all the help and feedback.
